# Clinicopathological Significance of *RUNX1* in Non-Small Cell Lung Cancer

**DOI:** 10.3390/jcm9061694

**Published:** 2020-06-02

**Authors:** Yujin Kim, Bo Bin Lee, Dongho Kim, Sangwon Um, Eun Yoon Cho, Joungho Han, Young Mog Shim, Duk-Hwan Kim

**Affiliations:** 1Department of Molecular Cell Biology, Sungkyunkwan University School of Medicine, Suwon 440-746, Korea; yujin0328@hanmail.net (Y.K.); whitebini@hanmail.net (B.B.L.); jindonghao2001@hotmail.com (D.K.); 2Department of Internal Medicine, Samsung Medical Center, Sungkyunkwan University School of Medicine, Seoul 135-710, Korea; sangwon72.um@samsung.com; 3Department of Pathology, Samsung Medical Center, Sungkyunkwan University School of Medicine, Seoul 135-710, Korea; eunyoon.cho@samsung.com (E.Y.C.); joungho.han@samsung.com (J.H.); 4Department of Thoracic and Cardiovascular Surgery, Samsung Medical Center, Sungkyunkwan University, School of Medicine, Seoul 135-710, Korea; youngmog.shim@samsung.com

**Keywords:** lung cancer, RUNX1, methylation, biomarker, survival

## Abstract

This study aimed to understand the clinicopathological significance of runt-related transcription factor 1 (*RUNX1*) in non-small cell lung cancer (NSCLC). The methylation and mRNA levels of *RUNX1* in NSCLC were determined using the Infinium HumanMethylation450 BeadChip and the HumanHT-12 expression BeadChip. RUNX1 protein levels were analyzed using immunohistochemistry of formalin-fixed paraffin-embedded tissues from 409 NSCLC patients. Three CpGs (cg04228935, cg11498607, and cg05000748) in the CpG island of *RUNX1* showed significantly different methylation levels (Bonferroni corrected *p* < 0.05) between tumor and matched normal tissues obtained from 42 NSCLC patients. Methylation levels of the CpGs in the tumor tissues were inversely related to mRNA levels of *RUNX1*. A logistic regression model based on cg04228935 showed the best performance in predicting NSCLCs in a test dataset (N = 28) with the area under the receiver operating characteristic (ROC) curve (AUC) of 0.96 (95% confidence interval (CI) = 0.81–0.99). The expression of RUNX1 was reduced in 125 (31%) of 409 patients. Adenocarcinoma patients with reduced RUNX1 expression showed 1.97-fold (95% confidence interval = 1.16–3.44, *p* = 0.01) higher hazard ratio for death than those without. In conclusion, the present study suggests that abnormal methylation of *RUNX1* may be a valuable biomarker for detection of NSCLC regardless of race. And, reduced RUNX1 expression may be a prognostic indicator of poor overall survival in lung adenocarcinoma.

## 1. Introduction

Lung cancer is the leading cause of cancer-related death in the world. Despite significant advances in the diagnosis and treatment of the disease over the past 20 years, its prognosis is still very poor, with the overall 5-year survival rate staying at 15%–20% [1]. The prognosis of cancer patients is mostly determined by disease stage. The occult metastatic spread of cancer cells to surrounding tissues in more than 50% of lung cancer patients at the time of diagnosis affects a poor prognosis. The majority of patients undergoing curative surgical resection at an early stage and, if necessary, adjuvant chemotherapy have achieved favorable long-term survival. Patients with surgically resected stage IA, stage IB, and stage II non-small cell lung cancers (NSCLCs) had an overall five-year survival rate of 83%, 69%, and 48%, respectively [2]. Targeted therapy has a great effect on the prognosis of specific patients; however, it is applicable to only about 10%–20% of patients. Accordingly, it is important to identify novel diagnostic and therapeutic biomarkers for the detection of early-stage lung cancer and for the development of new molecular-targeted therapies for NSCLC.

Runt-related transcription factor 1 (RUNX1) is one of the RUNX family proteins (RUNX1, RUNX2, and RUNX3), which forms a heterodimeric complex with the core binding factor β (CBFβ), resulting in enhanced transcription of the RUNX gene family by stimulating the DNA binding ability and stability of the family proteins [3,4]. RUNX1 is essential for hematopoiesis and is involved in the generation of hematopoietic stem cells. Mutations and translocations in *RUNX1* are well established as causes of myelodysplastic syndrome or acute myelogenous leukemia [5,6,7]. Gain- or loss-of-function mutations of *RUNX1* have also been reported in various solid tumors. Missense mutations of *RUNX1* were reported in luminal-type breast cancer [8], and loss-of-function somatic mutations or deletion of *RUNX1* have been reported in breast cancer and lung cancer [9,10].

To understand the clinicopathological significance of *RUNX1* in NSCLC, we analyzed the methylation status of *RUNX1* in different types of samples from a total of 118 NSCLC patients and 60 healthy individuals. The prediction performance of classifiers was validated in The Cancer Genome Atlas (TCGA) lung cancer. Expression levels of RUNX1 were also analyzed using HT-12 array and immunohistochemistry in tissue specimens from 42 and 409 NSCLC patients, respectively.

## 2. Materials and Methods

### 2.1. Study Population

Formalin-fixed paraffin-embedded tumor tissues were obtained from 409 NSCLC patients who underwent curative surgical resection at the Department of Thoracic and Cardiovascular Surgery, Samsung Medical Center, Seoul, Korea, between August 1994 and May 2014. All samples were obtained from operative patients. Follow-up of patients for the detection of recurrence or death following curative resection was conducted by a nurse specialized in oncology as described previously [11]. The pathological stage of NSCLC was determined using the tumor/node/metastasis (TNM) system provided by the American Joint Committee on Cancer (AJCC) [12]. This study was conducted in accordance with the ethical principles stated in the Declaration of Helsinki and approved by the Institutional Review Board (IRB#: 2010-07-204) of the Samsung Medical Center. Written informed consent to use pathological specimens for research was obtained from all patients prior to surgery. All data are not publicly available due to privacy and ethical restriction.

### 2.2. Analysis of RUNX1 Methylation and mRNA Levels

We previously analyzed the DNA methylation and mRNA expression at the level of genome using the Infinium HumanMethylation450 BeadChip and the HumanHT-12 expression BeadChips (Illumina, San Diego, CA, USA), respectively, in 42 surgically resected tumor and matched normal tissues, 136 bronchial washings, 12 sputums, or 6 bronchial biopsy specimens obtained from a total of 118 NSCLC patients and 60 cancer-free patients [13]. We used the reported data for the analysis of methylation and mRNA levels of *RUNX1*. Preprocessing such as background or batch effect correction, probe filtering, and adjustment of the background signal difference between types I and II probes was conducted using the R software package called wateRmelon [14]. Methylation level (β-value), ranging from 0 (no methylation) to 1 (100% methylation), was estimated as the ratio of fluorescence signal intensity between methylated alleles and the sum of methylated and unmethylated alleles at each CpG locus. The levels of mRNA expression from HT-12 chips were normalized using the R lumi package (https://bioconductor.org/biocLite.R).

### 2.3. Feature Selection for Prediction of Lung Cancer

To select candidate CpGs for lung cancer prediction among differentially methylated CpGs and to build models for lung cancer prediction, we divided the normal and tumor tissues from 42 patients into training and test datasets, according to a 7:3 ratio. Supervised machine learning algorithms were applied to select features in the training dataset. Age-related CpGs or any CpGs that were significantly correlated in the normal or tumor tissues were removed during the model building. Supervised machine learning algorithms for feature selection and model building were applied using RapidMiner Studio version 8.2 (RapidMiner Inc, Boston, MA, USA).

### 2.4. Evaluation of Prediction Performance of Models in The Cancer Genome Atlas (TCGA) Lung Cancer

Prediction performance of selected models was further tested using 899 TCGA lung cancers, including 75 normal and 824 tumor tissues. The performance was tested without distinction between adenocarcinoma and squamous cell carcinoma. The prediction performance of models was evaluated using receiver operating characteristic (ROC) curves, plotted using the MedCalc statistical Software version 19.0.5 (MedCalc Softward bvba, Ostend, Belgium).

### 2.5. Immunohistochemistry

The expression of RUNX1, Ki-67, phospho-pRb (Ser-807/811) proteins in the 409 NSCLC patients was determined using immunohistochemistry of tissue microarrays (TMAs). In brief, the 4-mm–thick TMA tissue sections on glass slide were deparaffinized in xylene and rehydrated in a series of decreasing concentrations of alcohol. Antigens were recovered by putting sections into 10 mmol/L citrate buffer solution (pH 6.0) and by heating in a microwave oven for 10 min. The sections were then incubated overnight at 4 °C with a mouse monoclonal antibody to RUNX1 (AML1/RUNX1 Antibody (clone 3A1) IHC-plus™ LS-B5382, LifeSpan BioSciences, Seattle, WA, USA), a polyclonal anti-phospho-pRb (Ser-807/811) antibody (Cell Signaling, Danvers, MA, USA), and a mouse monoclonal anti-Ki-67 (DAKO; clone MIB-1) antibody. Immunoreactivity of the proteins was detected using the Envision-Plus/horseradish peroxidase system (Dako, Carpinteria, CA, USA), and the antibody-bound peroxidase activity was visualized by incubating in 0.05% 3,3′-diaminobenzidine tetrahydrochloride (DAB) for 3 min at room temperature. All sections were counterstained with Mayer’s hematoxylin and a negative control was included by excluding the primary antibody each time. Three samples with RUNX1 expression in normal bronchial epithelial cells were used as a positive control for RUNX1 staining, and IHC was performed in duplicate.

### 2.6. Interpretation of Immunohistochemical Staining

The immunohistochemical stainings were interpreted by two authors (EY Cho and D-H Kim) in a double-blinded fashion, and samples showing poor inter-rater reliability (κ < 0.20) were removed from data analysis. RUNX1 expression was considered positive when nuclear staining was present, and the intensity and proportion of positive nuclear staining was assessed for scoring. A composite score of RUNX1 protein expression was semi-quantitatively calculated by multiplying the proportion score of positive cells (0, absent; 1, 0%–10%; 2, 10%–50%; 3, 50%–80%; and 4, >80%) with the staining intensity score (0, none; 1, weak; 2, moderate; and 3, strong). A cutoff for the reduced expression of RUNX1 protein was determined by taking into account the distribution of the composite scores between normal and tumor tissues and by comparing both false-negative and false-positive rates at different cutoffs. RUNX1 expression was considered reduced in a tumor with a composite score less than two. To score for phospho-pRb (Ser-807/811) and Ki-67, positive staining was determined according to the percentage of positively stained nuclei.

### 2.7. Statistical Analysis

Univariate analysis was performed using the *t*-test (or Wilcoxon rank sum test) and the chi-square test (or Fisher’s exact test) for continuous and categorical variables, respectively. The correlation between methylation levels of CpGs in the *RUNX1* gene was analyzed using Spearman’s rank correlation coefficient. The effect of reduced RUNX1 expression on survival was estimated using the Kaplan–Meier survival curves, and the difference between the survival curves of any two groups was evaluated by the log-rank test. Cox proportional hazards analysis was conducted to estimate the hazard ratios of reduced RUNX1 expression for survival after controlling for potential confounding factors. Statistical analysis was conducted using R software version 3.3.3. (R Foundation for Statistical Computing, Vienna, Austria).

## 3. Results

### 3.1. RUNX1 Hypermethylation Is Inversely Associated with Its Expression

Data reported previously were used to identify differentially methylated CpGs in *RUNX1* gene in tumor and matched normal tissues from 42 NSCLC patients. The Wilcoxon rank-sum test was applied because the distribution of β-values obtained from tumor tissues using a 450 K array was negatively skewed and did not follow a normal distribution (Shapiro-Wilk test, *p* < 0.05). Three CpGs with a *p*-value less than or equal to 1.03 × 10^−7^ (Bonferroni significance threshold) were identified from the 450K array: three CpGs (cg11498607, cg04228935, cg05000748) at the CpG island of *RUNX1* showed hypermethylation in tumor tissues compared with normal tissues (Figure 1A). The methylation levels of the three CpGs did not vary significantly with histology (Figure 1B). Altered methylation of three CpGs was not significantly correlated with a patient’s age (Figure 1C).The methylation levels were not also associated with smoking status (Figure 1D) and recurrence (Figure 1E). However, the methylation levels were found to be higher in the poorly differentiated type of NSCLC than in the well differentiated type (Appendix A). The *RUNX1* mRNA levels were analyzed using the HT-12 array to determine the association between methylation changes and changes in *RUNX1* gene expression. The methylation levels of individual CpGs were negatively associated with the mRNA levels of *RUNX1* (*p* < 0.05; Figure 1F).

### 3.2. Prediction of Non-Small Cell Lung Cancer (NSCLC) Using Abnormal Methylation Levels of RUNX1

Features for prediction of NSCLC were selected in 42 tumors and matched normal tissues. Lung tumor and matched normal tissues were divided into training and test datasets at a ratio of 7:3, respectively. We built models using the training dataset and tested the performance of the models using the test dataset. Supervised machine learning algorithms such as k-nearest neighbor (kNN), support vector machine (SVM), neural network, logistic regression, and decision tree were applied for feature selection. Since individual CpGs were correlated with each other, only one CpG was included in the models. Among the applied algorithms, a logistic regression model based on cg04228935 showed the best performance in classifying NSCLCs in a test dataset (N = 28) with a sensitivity of 92.9% and a specificity of 92.9% (area under the curve (AUC) = 0.96; 95% confidence interval (CI) = 0.81–0.99, *p* < 0.0001; Figure 2A).

To determine if *RUNX1* hypermethylation may be a biomarker for the detection of NSCLC in other races, we tested *RUNX1* hypermethylation in the 899 TCGA primary lung cancers (75 normal tissues and 824 tumor tissues). As with our data, the TCGA data was divided into a training dataset (N = 630) and a test dataset (N = 269), and the performance of logistic regression model based on three CpGs was evaluated on the test dataset (Appendix A). The sensitivity and specificity of the model based on cg04228935 in a test dataset (N = 269) were 91.8% and 96.4%, respectively. AUC was 0.95 (95% confidence interval = 0.93–0.98, *p* < 0.0001). The degree of prediction certainty of NSCLC in the test datasets was high in our data and TCGA lung cancer data (Figure 2B). We finally compared the methylation levels of three CpGs at a CpG island of *RUNX1* between our data and TCGA lung cancer data. No significant difference was found between the two data (Figure 2C).

### 3.3. Methylation Pattern of RUNX1 in Tumor Tissue Is Similar to that in Bronchial Biopsy Specimen

To test if bronchial washing, sputum, and bronchial biopsy specimens could be used as surrogate samples for analyzing *RUNX1* methylation in the lung, we compared the methylation levels of the CpG (cg04228935) in 42 tumors and matched normal tissues, 136 bronchial washings, 12 sputum samples, and 6 bronchial biopsy specimens. The clinical and pathological characteristics of the NSCLC patients were previously reported [13]. The methylation levels of the CpG in lung tumor tissues were not significantly different from those in bronchial biopsy specimens from lung cancer patients (*p* > 0.05, Wilcoxon rank sum test), unlike bronchial washings and sputum samples (Figure 3A). The CpG methylation levels were further compared between paired bronchial washing and sputum samples from 12 NSCLC patients (Figure 3B) and between bronchial washing and paired bronchial biopsies from 6 NSCLC patients (Figure 3C). The methylation levels of the CpG were found to be similar between bronchial washings and sputum samples but significantly higher in bronchial biopsy than in bronchial washing (*p* < 0.05, Wilcoxon signed-rank test). These findings suggest that a detection model for NSCLC using abnormal methylation of *RUNX1* is applicable to bronchial biopsy specimens.

### 3.4. RUNX1 Affects Overall Survival in Adenocarcinoma

To elucidate the effect of RUNX1 expression on survival of NSCLC patient, we analyzed the expression of RUNX1 using immunohistochemistry of formalin-fixed paraffin-embedded tumor tissues from 409 NSCLCs. Representative positive staining patterns of RUNX1 are shown in adenocarcinoma and squamous cell carcinoma (Figure 4A). Clinicopathological characteristics of 409 non-small cell lung cancer patients are listed in Appendix A. The median follow-up period of patients was 5.2 years. RUNX1 expression was reduced in 31% of the samples. Reduced RUNX1 expression was not related to pathologic stage (Appendix A), but was found more frequently in woman (Figure 4B) and in adenocarcinoma (Figure 4C) and was significantly associated with poor overall survival in adenocarcinoma (*p* = 0.005; Figure 4D) but not in squamous cell carcinoma (*p* = 0.87). The median survival of adenocarcinoma patients with and without reduced RUNX1 expression was 41 and 81 months, respectively. Cox proportional hazards analysis also showed that overall survival of adenocarcinoma patients with reduced RUNX1 expression was approximately 1.97 (95% CI = 1.16–3.44; *p* = 0.01) times poorer than in those without, after controlling for age, recurrence, and pathologic stage (Table 1). However, RUNX1 expression was not associated with recurrence-free survival irrespective of histology in NSCLC (*p* = 0.21).

### 3.5. No Correlation between Reduced RUNX1 Expression and Expression Levels of Phospho-Rb and Ki67 Proliferation Index

RUNX proteins are implicated in diverse signaling pathways and cellular processes, including the cell cycle and stress response. In order to elucidate the effect of RUNX1 on the cell cycle and cell proliferation in NSCLC, we analyzed the phospho-pRb (Ser-807/811) levels and Ki-67 proliferation index according to the expression status of RUNX1. The average phospho-pRb (Ser-807/811) levels were 2.7% in tumor tissues with reduced RUNX1 expression and 2.1% in tumor tissues without reduced RUNX1 expression. The difference was not statistically significant (*p* = 0.18), irrespective of histology (Figure 5A). The Ki-67 proliferation index in tumor tissues with reduced RUNX1 expression was slightly higher than in tumor tissues without reduced RUNX1 expression, but the difference was also not statistically significant (28.8% vs. 22.3%, *p* = 0.18), irrespective of histology (Figure 5B).

## 4. Discussion

RUNX1 causes a wide range of leukemias through translocation with genes such as eight-twenty-one (ETO) [7] and acts as an oncogene in various solid tumors such as ovarian cancer [15], and endometrial cancer [16], as well as in the mouse mammary tumor virus-polyoma middle tumor-antigen (MMTV-PyMT) transgenic mouse model of breast cancer [17], and in the transgenic adenocarcinoma of mouse prostate (TRAMP) model of prostate cancer [18]. RUNX1 is also known to function as a tumor suppressor in different types of cancer. For example, the ectopic expression of RUNX1 in esophageal adenocarcinoma cells reduced the anchorage-independent growth [19], and the knockdown of *RUNX1* by siRNAs enhanced androgen-independent proliferation of prostate cancer cells [20]. In addition, the inhibition of endogenous *RUNX1* using short-hairpin RNA targeting RUNX1 (shRunx1) in breast cancer cells resulted in loss of epithelial morphology and promotion of epithelial-mesenchymal transition [9], and the ectopic expression of RUNX1 reduced the population of breast cancer stem cells [21]. RUNX1 also inhibited the migration and stemness of mammary epithelial cells [22]. Ramsay et al. [10] reported that lentiviral-mediated RNAi knockdown of *RUNX1* increased the proliferation and migration of lung cancer cells. In the present study, *RUNX1* showed abnormal methylation in primary NSCLCs, and the reduced expression of RUNX1 was associated with poor overall survival, suggesting that RUNX1 may play a role as a tumor suppressor in normal bronchial epithelial cells.

Functional disruption of RUNX1 usually occurs by chromosomal translocation, point mutation, or deletion in leukemia and some solid tumors. *RUNX1* mutation has been reported rarely in lung cancer [23], although changes in its methylation have been reported by a couple of studies [24,25]. In this study, *RUNX1* was found to be abnormally methylated at the CpG island of *RUNX1* in NSCLC tumor tissues, and the methylation and mRNA levels of *RUNX1* showed a linear negative correlation. Unlike most genes whose transcription is regulated by a single promoter, *RUNX1* is regulated by two promoters in the upstream region of 5′ UTR [26]. The three hypermethylated CpGs in this study might affect the transcription of *RUNX1*, which may also be affected by tissue-specific control factors. Further studies are needed to understand the mechanisms underlying transcriptional repression mediated by abnormal methylation of *RUNX1* in NSCLC.

A CpG (cg04228935) for the prediction of NSCLC was identified using tumor and matched normal tissues obtained from 42 NSCLC patients. Although the number of normal samples in the TCGA lung cancer data is small and the prevalence of lung cancer is not the exact same between Koreans and Americans, the present study suggests that RUNX1 hypermethylation may be a useful biomarker for the early detection of NSLC in other populations worldwide. Screening of lung cancer using low-dose computed tomography (LDCT) reduces mortality; however, approximately 20% of pulmonary nodules were found to be false positive [27,28]. A biopsy is needed for a more accurate diagnosis of lung cancer, but it is very difficult to obtain tissue in some patients. Methylation levels of a CpG (cg04228935) from bronchial biopsy were comparable to those from surgically resected lung tumor tissues. Accordingly, bronchial biopsy specimens may be used for the molecular analysis of *RUNX1*, and advances in technology such as electromagnetic navigation bronchoscopy (ENB) and endobronchial ultrasonography using a guided sheath (EBUS-GS) may provide more adequate specimens with fewer complications.

The association of *RUNX1* mutations or changes in expression with the prognosis of patients has been reported in various carcinomas, and the effect of RUNX1 on prognosis varies considerably depending on the type of cancer. *RUNX1* mutations are associated with poor overall survival in adult acute myelogenous leukemia (AML) as well as in pediatric AML [29,30]. The RUNX1 expression in prostate cancer tissues was negatively associated with poor prognosis [20]. The low RUNX1 expression in breast cancers is associated with metastasis to lymph nodes and poor survival [9,21]. In addition, the RUNX1-RUNX3 expression showed a significant effect on the survival of breast cancer patients with high YAP-signature expression levels [22]. Lung adenocarcinomas with low RUNX1 expression were associated with poor overall survival compared to tumors with high RUNX1 expression [10]. Our data also showed that reduced RUNX1 expression was associated with poor overall survival in adenocarcinomas. Based on these observations, it is likely that the reduced expression of RUNX1 may serve as an indicator of poor prognosis in patients with lung adenocarcinoma.

RUNX proteins are known to regulate a wide range of biological processes via various interacting proteins in human cancer and to be implicated in carcinogenesis mediated via TGF-β and Wnt signaling pathways, and in cell cycle or stress response. For example, RUNX1 promoter is regulated by EZH2 (enhancer of zeste homolog 2)-dependent histone H3 lysine 27 (K27) trimethylation in prostate cancer cells [20]. RUNX1 directly regulates E-cadherin, and rescues TGFβ-induced EMT phenotype in breast cancer cells [9]. RUNX1 suppresses breast cancer growth by repressing the activity of breast cancer stem cells and inhibiting ZEB1 expression directly [21]. RUNX1 acts as a negative regulator of oncogenic function of YAP that is involved in solid tumor progression [22]. The effect of RUNX1 on cell cycle in lung cancer differs between study groups. RUNX1 stimulated G1 to S progression in hematopoietic cells, partly via transcriptional induction of cyclin D2 promoter [31], whereas RUNX1 depletion resulted in an increased E2F1 mRNA levels in lung cancer cells [10]. In this study, tumor tissues with reduced RUNX1 expression did not show high levels of pRb phosphorylation (Ser-807/811) or the Ki67 proliferation index, suggesting that the reduced expression of RUNX1 may be involved in lung carcinogenesis through other mechanisms rather than cell-cycle regulation and growth control.

This study was limited by several factors. First, the effect of the two promoters on abnormal methylation of three CpGs in *RUNX1* gene and the tissue-specific factors affecting the expression of RUNX1 were not fully elucidated. Second, we failed to analyze *RUNX1* methylation in circulating cell-free DNA and to evaluate the prediction performance of the model due to assay failure. Third, the present study was a retrospective case-control study, which can result in a biased estimate of the population prevalence of NSCLC. In addition, sputum analysis was limited to the very few specimens from NSCLC patients only. Accordingly, the prediction performance of the model needs to be validated using several molecular techniques such as droplet digital polymerase chain reaction (ddPCR) in sputums and cell-free DNAs from a large cohort. Fourth, it is unclear the abnormal methylation of RUNX1 as a predictive biomarker can also be applied to tissue samples from metastatic lesions because the data from the present study and TCGA was from surgical specimens of early stage tumors. Fifth, methylation levels from benign lung tumors such as localized organizing pneumonia and hamartoma were not analyzed due to lack of samples.

In conclusion, the present study suggests that abnormal methylation at the CpG island of the *RUNX1* gene may be a valuable biomarker for the detection of NSCLC regardless of races. Reduced expression of RUNX1 may be associated with poor overall survival in patients with lung adenicarcinoma.

## Figures and Tables

**Figure 1 jcm-09-01694-f001:**
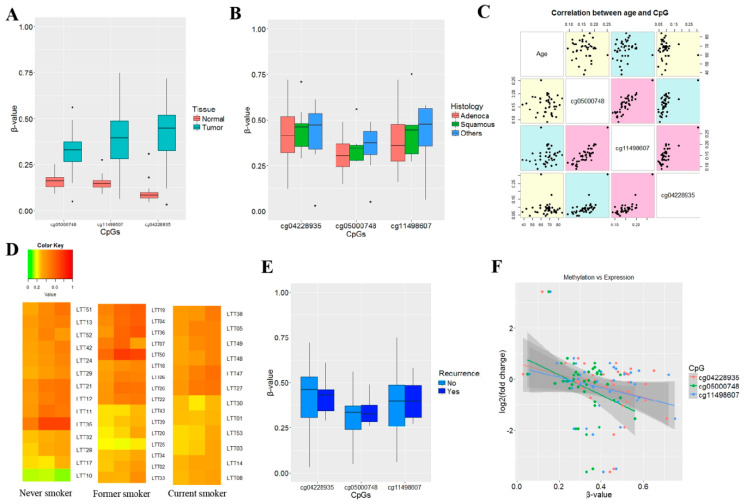
Relationship between methylation and mRNA levels of runt-related transcription factor 1 (*RUNX1**)* in 42 lung tumor and matched normal tissues. (**A**) Methylation levels of three CpGs at the CpG island of *RUNX1* gene were compared between the tumor and matched normal tissues obtained from 42 NSCLC patients. Y-axis indicates β-values. (**B**) Methylation levels of the three CpGs were compared according to histologic subtypes. (**C**) Correlations between the patient’s age and the methylation levels of the three CpGs were analyzed in 42 tumor tissues. Spearman’s correlation coefficient was used to calculate *p*-values. Magenta color indicates *p* < 0.05. (**D**) Methylation levels of three CpGs at a CpG island of *RUNX1* were compared in never-smokers, former smokers, and current smokers. Y-axis indicates sample identification numbers. Methylation levels are represented using gradient-based colors from green (0%–20%) to yellow (21%–50%) to red (51%–100%). (**E**) The association between recurrence and the methylation levels at three CpGs were analyzed in 42 NSCLCs. (**F**) The correlation between methylation levels of three CpGs and the mRNA expression of *RUNX1* was analyzed in 42 tumor tissues from patients with NSCLC. Y-axis indicates the log2 fold change (= log2(tumor/normal)) between tumor and matched normal tissues. X-axis indicates β-values in tumor tissues.

**Figure 2 jcm-09-01694-f002:**
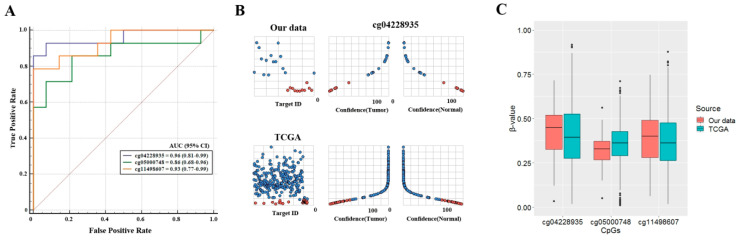
Evaluation of prediction performance of five supervised machine learning algorithms in non-small cell lung cancer (NSCLC). (**A**) The true and false positive rates of logistic regression model based on three CpGs were evaluated in a test dataset (N = 28) of 42 NSCLCs, and the receiver operating characteristic (ROC) curves were plotted using the MedCalc software. (**B**) The prediction certainty of the support vector machine model was evaluated in the test dataset of our data and TCGA lung cancer. The X-axis indicates the degree (0% to 100%) of certainty for prediction of our and TCGA tissues as normal or tumor for each β-value on the Y-axis. The sky blue and red orange circles indicate tumor and normal tissues, respectively. (**C**) The β-values of the three CpGs in our and TCGA data were compared to understand the difference of *RUNX1* hypermethylation among other ethnic groups or populations.

**Figure 3 jcm-09-01694-f003:**
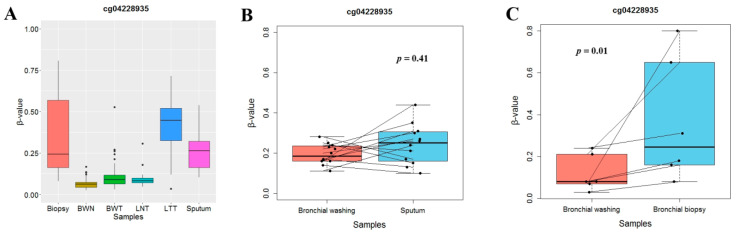
Comparison of *RUNX1* methylation levels among different types of specimens. (**A**) The methylation levels of a CpG (cg04228935) selected for NSCLC prediction were compared among bronchial biopsies from 6 lung cancer patients (biopsy), bronchial washing samples from 60 healthy individuals (bronchial washing normal, BWN) and 76 lung cancer patients (bronchial washing tumor, BWT), tumor (lung tumor tissue, LTT) and matched normal (lung normal tissue, LNT) tissues from 42 NSCLC patients, and sputum specimens from 12 lung cancer patients (sputum). (**B**,**C**) Methylation levels of a CpG (cg04228935) were compared using parallel coordinate plots between paired bronchial washing and sputum specimens from 12 NSCLC patients (**B**) and between paired bronchial washing and biopsy samples from six NSCLC patients (**C**). Methylation levels in bronchial washings were similar to those in sputum samples but were significantly low compared with those in bronchial biopsy (Wilcoxon signed-rank test). Y-axis indicates β-values from the 450 K array.

**Figure 4 jcm-09-01694-f004:**
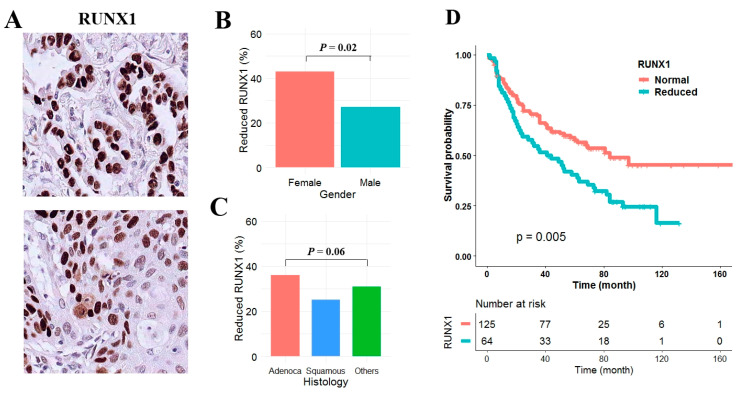
The effect of RUNX1 expression on overall survival in NSCLC. (**A**) RUNX1 expression was analyzed using immunohistochemistry in 409 NSCLC patients. Positive staining occurred in the nucleus of adenocarcinoma (upper) and squamous cell carcinoma. (X200). (**B**,**C**) Reduced expression levels were compared according to gender (**B**) and histologic subtypes (**C**). *p*-values are based on Student *t*-test. (**D**) The effect of reduced RUNX1 expression on overall survival of patients with adenocarcinoma was analyzed using Kaplan-Meier survival curve. *p*-value was calculated using log-rank test.

**Figure 5 jcm-09-01694-f005:**
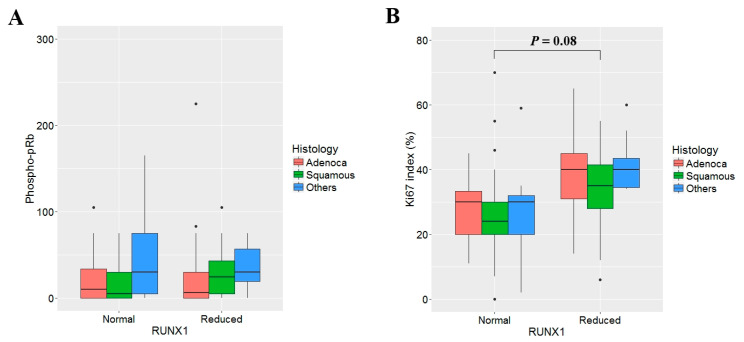
The effect of RUNX1 expression on phospho-pRb (Ser-807/811) level and Ki-67 proliferation index. The expression levels of phosphorylated pRb stained using polyclonal anti-phospho-pRb (Ser-807/811) antibody (Cell Signaling, Danvers, MA, USA) (**A**) and Ki-67 proliferation index (**B**) were compared according to the expression status of RUNX1 using Student *t*-test. The phospho-pRb levels and Ki-67 proliferation index were not significantly different between tumor tissues with normal or reduced RUNX1 expression irrespective of histologic subgroup.

**Table 1 jcm-09-01694-t001:** Cox proportional hazards analysis of overall survival according to RUNX1 expression.

Histology	RUNX1 Expression	HR	95% CI	*p*-Value
Adeno (N = 189)	Normal	1.00		
	Reduced	1.97	1.16–3.44	0.01
Squamous (N = 192)	Normal	1.00		
	Reduced	1.46	0.78–5.32	0.21

Abbreviations: Adeno, adenocarcinoma; squamous, squamous cell carcinoma; HR, hazard ratio; CI, confidence interval; RUNX1, runt-related transcription factor 1.

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
