# Peer review of "Clinicopathological Significance of RUNX1 in Non-Small Cell Lung Cancer"

_jcm, 2020, doi:10.3390/jcm9061694_

Round 1

Reviewer 1 Report

In the current manuscript, Kim at al describes a series of retrospective tissue analysis from early-stage surgical specimens of patients with NSCLC to evaluate the significance of methylation and mRNA levels of RUNX1 in NSCLC. The authors found that RUNX1 expression was reduced in 31% of patients. RUNX1 hypermethylation is inversely related to RUNX1 expression, and reduced RUNX1 expression was associated with poor overall survival in patients with adenocarcinoma.

Major revisions 

  • The authors mention in multiple areas in the manuscript and conclude that RUNX1 is a biomarker for early detection of NSCLC. In section 3.2 they describe algorithms for using RUNX1 for prediction of NSCLC. I am not sure if we can reach to the strong conclusion that RUNX1 can be used for detection of NSCLC from the results of the current manuscript. These results just establish a correlation, not causality.
  • Like TCGA, the data from the current manuscript is from surgical specimens of early-stage tumours. It is unclear the of RUNX1 as predictive biomarker would still hold in tissue samples from metastatic lesions. This should be mentioned as a limitation  
  • The authors mention that RUNX1 methylation was not correlated with patients age, smoking status or recurrence. Was there any correlation with differentiation ( well-differentiated vs poorly differentiated )

Minor

- Line 42-42. The authors mention that targeted therapy is applicable only in 10% patients but there is no reference mentioned here. I am not sure if this is accurate. EGFR mutations alone are present in 10- 20% of patients with NSCLC so this number would be way higher   

Author Response

Response to Reviewer 1 Comments

Point 1: Like TCGA, the data from the current manuscript is from surgical specimens of early-stage tumours. It is unclear the of RUNX1 as predictive biomarker would still hold in tissue samples from metastatic lesions. This should be mentioned as a limitation.

Response 1: As the reviewer pointed out, we have added this issue as a limitation in the Discussion section (Lines 355 - 358).

Point 2: The authors mention that RUNX1 methylation was not correlated with patients age, smoking status or recurrence. Was there any correlation with differentiation ( well-differentiated vs poorly differentiated )

Response 2: RUNX1 hypermethylation occurred more frequently in poorly differentiated type than in well-differentiated type. We have added this data to the Result (Supplementary Fig. S1) (Lines 155-156).

Point 3: The authors mention that targeted therapy is applicable only in 10% patients but there is no reference mentioned here. I am not sure if this is accurate. EGFR mutations alone are present in 10- 20% of patients with NSCLC so this number would be way higher.  

Response 3: As the reviewer pointed out, 10% is incorrect, Therefore, we have changed the number to 10-20% (Line 46).

Reviewer 2 Report

  1. According to the title of this manuscript, it could be expected that authors have proved that abnormal methylation of RUNX1 gene can be used as a diagnostic biomarker in the NSCLC. Unfortunately, presented results don’t point strongly to possibility of such application. It has been evidently shown that the methylation level of three analyzed CpGs was significantly different between lung cancer tissue and matched normal tissue obtained from 42 NSCLC patients. Based on this difference, the authors have built some models using machine learning methods for effective classifying cancerous and normal lung tissue. The accuracy of these tests was about 96%. Afterwards these models have been positively validated in The Cancer Genome Atlas (TCGA) lung cancer. The distinguishing between NSCLC and normal tissue seems to be without a doubt. However, taking into account the other data provided in this study, these models don’t fully meet criteria for biomarker definition.  
    First of all they can’t be used for screening and early diagnosis of NSCLC. The methylation levels of the CpG were found to be similar between bronchial washings from healthy individuals and lung cancer patients; sputum analysis was unfortunately limited to the specimen from NSCLC patients only. Measuring of the CpG’s methylation level in specimen biopsied from visible, pathological foci in lungs or bronchi could be only an additional method without providing any other, relevant for diagnosis, details in most cases 
    The next serious flaw in this research has been  a lack of methylation level assessment in other benign and malignant conditions which have to be included in the differential diagnosis of NSCLC. 
  2. The presented manuscript contains results of several experiments and analysis. However, only some of them cover the main subject. For example, the results of RUNX1 expression, its clinical significance and its impact on the cell cycle regulation and growth control in NSCLC, are not strictly related with establishing of CpGs methylation as diagnostic biomarker. To sum up, this paper should be dedicated rather to the RUNX1 role evaluation and its expression control in non-small cell lung cancer. In my opinion the new title and main aims redefinition could improve the scientific soundness of this manuscript. 
  3. The authors have also postulated the reduced RUNX1 expression as a negative prognostic factor for overall survival in patients with adenocarcinoma. I would personally be more careful in the interpretation of presented data this way: 
    - the treatment results of adenocarcinoma are highly linked to surgery. Unfortunately, initial status of tumors (operative vs inoperable) hasn’t been provided in the both analyzed cohorts (with and without reduced RUNX1 expression). 
    - a lack of association between RUNX1 expression and recurrence-free survival irrespective of histology may point to other reason of deaths in adenocarcinoma patients than RUNX1 expression 
    Furthermore, providing the incidence rate of RUNX1 reduced expression among particular stages and histologic types would be valuable 
  4. How many NSCLC patientswere finally included inanalysis of RUNX1 methylation level? The authors have stated: 
    - line 58: “[…] from a total of 141 NSCLC patients” 
    - line 79: “[…] obtained from a total 183 NSCLC patients” 
    - ref [13] (the source of data for this analysis) – “[..] A total of 118 lung cancer patients” 
    Moreover, the clinical and pathological characteristics of the patients included into this part of the study should be provided 

Author Response

Response to Reviewer 2 Comments

Point 1: According to the title of this manuscript, it could be expected that authors have proved that abnormal methylation of RUNX1 gene can be used as a diagnostic biomarker in the NSCLC. Unfortunately, presented results don’t point strongly to possibility of such application. It has been evidently shown that the methylation level of three analyzed CpGs was significantly different between lung cancer tissue and matched normal tissue obtained from 42 NSCLC patients. Based on this difference, the authors have built some models using machine learning methods for effective classifying cancerous and normal lung tissue. The accuracy of these tests was about 96%. Afterwards these models have been positively validated in The Cancer Genome Atlas (TCGA) lung cancer. The distinguishing between NSCLC and normal tissue seems to be without a doubt. However, taking into account the other data provided in this study, these models don’t fully meet criteria for biomarker definition.  
First of all they can’t be used for screening and early diagnosis of NSCLC. The methylation levels of the CpG were found to be similar between bronchial washings from healthy individuals and lung cancer patients; sputum analysis was unfortunately limited to the specimen from NSCLC patients only. Measuring of the CpG’s methylation level in specimen biopsied from visible, pathological foci in lungs or bronchi could be only an additional method without providing any other, relevant for diagnosis, details in most cases.  

Response 1: We concur with the reviewer that the present study did not support the significance of RUNX1 methylation as a biomarker for screening and early diagnosis of NSCLC. As the reviewer noted, sputum analysis was performed in very few specimens from NSCLC patients only. In addition, we failed to analyze RUNX1 methylation in cell-free DNAs from 120 NSCLCs and 120 controls as pyrosequencing did not work. However, the cg04228935 discovered in this study showed very high performance in predicting NSCLC compared to any other CpGs in different genes reported thus far. Accordingly, we believe that RUNX1 methylation should be further evaluated using other molecular techniques such as droplet digital PCR(ddPCR) in sputum and cell-free DNA of a large cohort. We have explained this issue in the Discussion (Lines 352-355).

Point 2: The next serious flaw in this research has been  a lack of methylation level assessment in other benign and malignant conditions which have to be included in the differential diagnosis of NSCLC. 

Response 2: We need certain methylation data from benign lung tumors such as localized organizing pneumonia and hamartoma for differential diagnosis of NSCLC. However, we can’t analyze benign lesion due to lack of the samples. We have explained this issue in the Discussion (Lines 358-359).

Point 3:The presented manuscript contains results of several experiments and analysis. However, only some of them cover the main subject. For example, the results of RUNX1 expression, its clinical significance and its impact on the cell cycle regulation and growth control in NSCLC, are not strictly related with establishing of CpGs methylation as a diagnostic biomarker. To sum up, this paper should be dedicated rather to the RUNX1 role evaluation and its expression control in non-small cell lung cancer. In my opinion the new title and main aims redefinition could improve the scientific soundness of this manuscript. 

Response 3: We have changed the title and main aims (Lines 4-5, line 59).

Point 4: The authors have also postulated the reduced RUNX1 expression as a negative prognostic factor for overall survival in patients with adenocarcinoma. I would personally be more careful in the interpretation of presented data this way: the treatment results of adenocarcinoma are highly linked to a surgery. Unfortunately, initial status of tumors (operative vs inoperable) hasn’t been provided in the both analyzed cohorts (with and without reduced RUNX1 expression). 

Repsonse 4: We have explained that all samples were obtained from operative patients in the Materials and Methods (Lines 69-70).

Point 5: a lack of association between RUNX1 expression and recurrence-free survival irrespective of histology may point to other reason of deaths in adenocarcinoma patients than RUNX1 expression.  

Response 5: To reconfirm the issue, we have reanalyzed the adenocarcinoma patients including all possible confounding factors and obtained the following results.

Variables

HR

95% CI

P-value

RUNX1

1.97

1.16 - 3.44

0.01

Pathologic stage

2.10

1.59 - 2.76

<0.0001

Age

1.03

1.01 - 1.06

0.006

Recurrence

2.83

1.69 - 4.75

<0.0001

Therefore, we conclude that reduced RUNX1 expression may be an independent prognostic factor for overall survival in adenocarcinoma of the lung. We have changed the hazard ratio of RUNX1 in the Result (Lines 29-30, .lines 246-248, Table 1)

Point 6: Furthermore, providing the incidence rate of RUNX1 reduced expression among particular stages and histologic types would be valuable  

Response 6: We have added a figure (Supplementary Fig. S2) that shows the incidence rate of reduced RUNX1 expression according to pathologic stages (Line 241). The relationship of RUNX1 expression on histologic subtypes is shown in Fig. 4C.

Point 7: How many NSCLC patients were finally included in analysis of RUNX1 methylation level? The authors have stated: 
- line 58: “[…] from a total of 141 NSCLC patients” 
- line 79: “[…] obtained from a total 183 NSCLC patients” 
- ref [13] (the source of data for this analysis) – “[..] A total of 118 lung cancer patients” 
Moreover, the clinical and pathological characteristics of the patients included into this part of the study should be provided 

Response 7: In this manuscript, we showed the total number (N=141) of patients participating in the 450K array experiment. However, the number (N=118) shown in the source data (ref. 13) was limited to patients used for final data analysis. The 183 was a typo. In order to avoid readers’ confusion, we have unified the number of patients to 118(Line 61, lines 82-83). Clinical and pathological characteristics of the patients were previously reported in the reference. Therefore, we have introduced the reference in the Result (Lines 211-212).

Round 2

Reviewer 2 Report

I recommend this paper for publication.